# Factorial Structure of Trunk Motor Qualities and Their Association with Explosive Movement Performance in Young Footballers

**DOI:** 10.3390/sports9050067

**Published:** 2021-05-18

**Authors:** Jasminka Tomčić, Nejc Šarabon, Goran Marković

**Affiliations:** 1Faculty of Kinesiology, University of Zagreb, 10110 Zagreb, Croatia; aamika@gmail.com; 2Dorsalis, Ltd., 10000 Zagreb, Croatia; 3Laboratory for Motor Control and Motor Behavior, S2P, Science to Practice, Ltd., SI-1000 Ljubljana, Slovenia; nejc.sarabon@fvz.upr.si; 4Faculty of Health Sciences, University of Primorska, SI-6310 Izola, Slovenia; 5Motus Melior, Ltd., 1000 Zagreb, Croatia

**Keywords:** core function, movement performance, stability

## Abstract

This study examined the factorial structure of trunk motor qualities and their associations with explosive muscular performance of football players. Ninety-one young male football players (age: 15–21 years; body height: 1.78 ± 0.07 m; body mass: 70.3 ± 7.5 kg) performed a series of tests: four standing and four seated isometric trunk strength tests, seven trunk power (medicine ball throwing) tests, four trunk endurance tests and four explosive movement performance tests. A principal component factor analysis (PCA) was used to determine the structure of trunk motor qualities, and correlational analyses were used to assess linear associations between trunk motor qualities and explosive performance. The PCA revealed four independent factors—trunk power, standing and seated isometric trunk strength, and trunk muscle endurance. Only trunk power had significant moderate, logically positive associations with sprint and vertical jump performance (common variance: 25–36%), while other associations between trunk motor factors and explosive movement performance were generally low to very low. These results indicate that trunk muscle functions of football players can be described with three independent motor qualities—trunk power, trunk strength and trunk muscle endurance, with only trunk power being moderately associated with players’ sprinting and jumping performance.

## 1. Introduction

The trunk is the central part of the human body and in the musculoskeletal sense includes the spine, pelvis, hip proximal parts of the lower extremities and abdominal structure [1]. In a narrower sense, the term core refers to the muscular assembly consisting of the abdominal muscles at the front and sides, the paraspinal and gluteal muscles at the back, the diaphragm at the top and at the bottom the pelvic floor muscles and the muscle complex around the hips [2]. Fig [3] defines the core as the area between the sternum and the knee with a focus on the abdominal muscles, lower back and hips. Authors who have studied the trunk or core in the context of sports performance commonly include a wider area emphasizing the importance of the shoulder and pelvis [4].

Given their central location in almost all functional kinetic chains, trunk muscles are responsible for the stability of the spine and pelvis and help generate and transfer force from large to smaller body parts providing proximal stability for distal mobility [1]. The trunk muscles have a dual role: stabilizing (static) and propulsive (dynamic) control [5]. Accordingly, the trunk has an important and unavoidable role in human movement in general and especially in sports activities, both in terms of impact on performance and injury prevention. Reduced trunk muscle function can affect the occurrence of lower and upper extremity injuries in athletes [6], but it is unclear to what extent the reduced trunk neuromuscular functions contribute to injury risk relative to other factors [5].

The current literature is ambiguous regarding distinguishing terms related to trunk motor qualities and in establishing their clear definitions. Often, different trunk motor qualities are reduced to a single one, i.e., trunk stability, but there is no consensus regarding its definition [4]. Thus, various concepts can be found under the term of trunk stability, such as hip and trunk muscle strength, trunk muscle endurance, maintaining a certain degree of pelvic inclination or spine position, and others. Silfies et al. [5] define trunk stability as “a dynamic process that requires optimal muscle capacity (power, endurance, strength) and neuromuscular control (appropriate joint and muscle receptors as well as neurological pathways) that can quickly integrate sensory information and change motor responses according to internal and external information”. According to Borghuis et al. [7], trunk stability involves the ability of the neuromuscular system to keep the trunk in an upright position or return it to that position after perturbation and to control the movements of the trunk itself during dynamic movements. In accordance with the problem of defining and differentiating the basic functions of the trunk muscles, we encounter the problem of their assessment and their interdependence. Often the authors in the same paper name one trunk function by different terms and measuring one trunk function they conclude about another [8,9,10,11]. Kibler et al. [1], consider that trunk functions and dysfunctions should be assessed through three-dimensional movements within which trunk muscles perform their functions.

In addition to the problem of structure and definition, there is a lack of evidence regarding the role of trunk motor qualities in sports performance. A recent meta-analysis by Prieske et al. [12] showed that the relationship between trunk strength and athletic performance is very low (*r* = −0.05 to 0.18). The same meta-analysis showed that large changes in trunk muscle strength, caused by trunk strength training, lead to small changes in the performance. Moreover, these training effects were smaller than the training effects induced by the standard whole-body resistance training. However, the authors point out the low level of methodological quality of the analyzed studies, which limits the generalization of these conclusions. A similar conclusion was reached by Reed et al. [13], after a detailed systematic review of the literature.

In order to address the above-mentioned issues related to structure of trunk motor qualities and their association with athletic performance, we have designed a cross-sectional study where we measured a relatively large group of athletes with a number of tests that hypothetically measure different trunk motor qualities, and we applied a principal component factor analysis (PCA) to define few, relatively independent trunk motor dimensions or factors. We have also analyzed the linear association between obtained trunk motor factors and athletic movement performance. Based on results of previous research, we hypothesized [1] that PCA will result in extraction of three relatively independent trunk motor qualities: strength, power, and muscular endurance, and [2] that the linear associations between these factors and athletic performance will be low to moderate.

## 2. Materials and Methods

### 2.1. Participants

Ninety-one healthy young male football players, all members of the football club GNK Dinamo Zagreb (age: 17.3 ± 1.7 years; age range: 15–21 years; body height: 1.78 ± 0.07 m; body mass: 70.3 ± 7.5 kg) participated in this cross-sectional study. Participants had at least 7 years of experience in football competition and were healthy at the time of the experiment. Their typical in-season weekly schedule included 5 football training sessions, 2 conditioning training sessions and one official match. The research was conducted according to Helsinki Declaration. Adult players confirmed their participation in the research with written consent, and for underage players their parents gave their consent. The Science and Ethics Committee of the Faculty of Kinesiology, University of Zagreb approved the experimental protocol (No. 7/17).

### 2.2. Test Procedure

All tests were conducted in the afternoon and were spread over three sessions, with a 48-h interval between testing sessions. Participants were asked not to participate in intensive training or competition activities the day before and on the day of testing. At the beginning of each testing session, a standard warm-up of 15 to 20 min was performed. It included (I) joint mobility exercises, (II) low intensity jogging combined with various dynamic tasks—squats, push-ups, sit-ups, prone spine extensions, lunges, planks, hip thrusts, and (III) dynamic stretching. The first testing session included isometric dynamometry of the trunk. The second testing session included medicine ball throws (i.e., trunk and upper-body explosive power tests) and trunk muscle endurance tests. In the third session, participants performed rapid movement performance tests. Within each session, the pause between tests was about 5 min.

### 2.3. Isometric Dynamometry of Trunk Muscles

Testing of the trunk muscles isometric strength was performed using a dynamometer (TNC, S2P Ltd., Ljubljana, Slovenia) with a built-in force sensor (PW10AC3-200 kg; HMB, Darmstadt, Germany). Given that the trunk muscle strength depends on the magnitude of flexion of the hip joint [14], isometric trunk strength testing was performed in both standing and sitting positions. Specifically, four trunk muscle groups were tested isometrically in both body positions: trunk flexors and extensors, as well as left and right trunk lateral flexors.

Standing position. All isometric trunk muscle tests were performed using a recently described testing procedure [15], (see also Figure 1). In brief, each subject took an upright position with feet placed shoulder-width apart and arms crossed over the chest (except in the case of flexion when the arms are placed in front of the trunk with the forearms bent and vertical to the floor). The pelvis was fixed by a strap to the lower support of dynamometer, the upper edge of which is placed at the height of the anterior superior iliac spine. The upper support of dynamometer, with a built-in force sensor, was positioned in a way that the upper edge of the support is leveled with the superior border of shoulder blades. The distance between the lower and upper support (in meters) was measured for each participant and was used for calculation of torque values. To acquire maximal voluntary contraction force (MVC), the participant was asked to press against the upper support as strongly as possible for 3 s. Three MVC trials (pause: 20–30 s) were acquired for pushing forward, backward, and aside (i.e., trunk flexion, extension, and lateral flexion). One practice trial was given to all participants.

Sitting position. Isometric trunk testing procedure in seated position (i.e., on a stable wooden box; hip flexion = 90°; see Figure 2) was similar to the one in standing position, except for the fact that the lower support of the dynamometer was placed at slightly higher position due to limitation imposed by participants’ thighs. As a result, the distance between the lower and upper support of the dynamometer was also measured in seated position.

For all isometric tests, the strongest among the three MVC trials (mean force on 1-s time interval) was used for further analysis. These results were multiplied by the length of the moment arm (in meters) and divided by participants’ body mass (in kilograms).

### 2.4. Explosive Power Tests (Medicine Ball Throws)

Explosive power tests were performed using 3 kg medicine ball throws proposed by Shinkle et al. [16] and Tse et al. [17]. To exclude the contribution of legs to throwing performance, medicine ball throws were performed in seated position, in four directions: forward, backward, and to the right and left side. Furthermore, each of the four throws were performed in two conditions with respect to the trunk stabilization—static and dynamic, i.e., with and without a back support. Each subject first performed static and then dynamic throws. Finally, a standing medicine ball forward throw was also performed. Collectively, a total of 9 explosive throwing tests were performed.

Medicine ball throws—sitting with back support (static trunk). Each participant took a sitting position on a weightlifting bench with raised backrest, with full feet resting on the floor and 90-degree of hip flexion. The back of a participant was upright throughout the throwing and the belt over the chest was firmly fixed to the backrest to prevent trunk movement. The throw was repeated if there was movement in the hip or trunk and if the feet rose from the ground. Each throw in 4 directions was performed 2 times, with a rest of 1 min between throws. In the further analysis, a better result of two throws per direction was used. The distance was measured in centimeters from the center of the seat to the point of first contact of the ball with the ground. (I) Static forward throw: the participant held the ball over his head with both hands, with abducted shoulders and flexed elbows, and then performed a forward throw. (II) Static backward throw: the participant held the medicine ball with both hands at navel height and then performed an overhead back throw. (III) Static side throw: the ball was held with both hands outside of the thigh, opposite to the throw side, and then performed the throw with minimal spinal rotation. The side throw was first performed to the right and then to the left. The stated order of all directions of throws with static trunk was the same for all participants.

Medicine ball throws—sitting without back support (dynamic trunk). Each participant took a sitting position on a weightlifting bench with lowered backrest, with full feet resting on the floor and 90-degree of hip flexion. If the feet rose from the floor during the throw, it was repeated. Each throw in 4 directions was performed 2 times, with a pause of 1 min between each throw. In the further analysis, a better result of two throws per direction was used. The distance was measured in centimeters from the center of the seat to the point of first contact of the ball with the ground. (I) Dynamic forward throw: the participant lifted the ball above the head with both hands, then, to achieve momentum, he leaned back as far as he wished, provided that his feet remained on the floor. The throw was performed by flexing hips and spine at the time of ball throw forward. (II) Dynamic backward throw: the participant leaned forward (hip and spine flexed) with the ball held with both hands in front of his knees; then he threw the ball backward overhead by extending his hips and trunk and raising his arms above his head. The participant remained seated throughout the throwing performance. (III) Dynamic side throw: The participant first flexed the hips and spine by rotating to the side opposite to the throwing side, and then, in preparation for throwing, increased flexion and rotation to achieve momentum and threw the ball to the opposite side. The dynamic side throw was first performed to the right and then to the left side by all subjects.

Standing forward medicine ball throw. The participant took standing position with feet hip-width apart and toes placed up to a line mark on the floor and then he lifted the ball above his head with both hands. In preparation for throw, the subject swung the ball by placing it behind his head, extended his hips and trunk and flexed his knees. Then he threw the ball by swinging his arms forward simultaneously with knee extension, and hip and trunk flexion. Stepping and any movement feet were not allowed. The throw was performed 2 times with a pause of 1 min between throws. In the further analysis, a better result of two throws was used. The distance was measured in centimeters from the line mark on the floor to the point of first contact of the ball with the ground.

### 2.5. Static Trunk Muscle Endurance Tests

Static endurance of trunk muscles was measured using a test battery proposed by McGill et al. [18]. Due to the high reliability of these tests (ICC > 0.97), each individual test was performed only once [19]. Extensors, flexors, and lateral flexors of the trunk on both sides were tested and a total of 4 variables were obtained that represent the dimensions of static endurance of the trunk muscles. The order of the tests was the same for all subjects and was performed according to the schedule: trunk extensors, trunk flexor, lateral flexors of right then left side. Static trunk muscle endurance tests were performed with at least 5 min rest interval between tests.

Trunk extensors endurance test—modified Biering–Sorensen test. The participant lay in a prone position on the therapy table in such a way that both anterior superior iliac spines were on the table, while the upper part of the trunk was off the table. The participant rested his hands on a box placed about 25 cm below the level of the table. The hips, knees and ankles were strapped to the table. The test began by lifting the trunk to a horizontal position at table level with the arms crossed over the chest and the palms placed on opposite shoulders. The participant was instructed to maintain a horizontal position for as long as possible and was verbally corrected if the horizontal position was disturbed. Time was measured in seconds with a hand-held stopwatch from the moment of taking a horizontal position until the moment when the participant’s upper body touched the box.

Trunk flexors endurance test—V-sit test. The participant was sitting on the floor, at a mark 10 cm away from a wooden backrest placed at an angle of 45° and his upper body resting on the backrest. His knees and hips were flexed at 90°, and his feet were firmly pressed to the ground. The feet were held and fixed to the floor by the assistant surveyor’s hands. The arms were crossed over the chest with the palms placed on opposite shoulders. The test began by moving the upper trunk away from the backrest by keeping the spine neutral and the trunk upright with the head and neck in a neutral position. The participant was instructed to maintain this position for as long as possible and was verbally corrected if he lost the neutrality of the spine, i.e., the upright position of the trunk. Time was measured in seconds with a hand stopwatch from the moment the back moved away from the backrest and the neutral position of the spine was taken until the moment when the participant’s upper body touched the backrest.

Trunk lateral flexors endurance test—side plank. The test was performed on a 2.5 cm thick exercise mat with a 5 cm wide self-adhesive tape set in the center along the entire mat. The participant took a lying position on a side and leaned on his forearm so that he placed his elbow (placed in the projection of his shoulder on the ground), hip and extended lower leg on the tape mark. The foot of the upper leg is placed in front of the lower leg foot in such a way that the heel of the upper foot touches the toes of the lower one. The free, upper arm is placed over the chest with the palm embracing the shoulder of the lower arm. The test begins by lifting the pelvis off the ground by leaning on the forearm and the edges of the foot and bringing the midline of the body into a straight line. The participant was instructed to maintain this position for as long as possible and was verbally corrected if he lost the straightness of the midline of the body and if he rotated around it. Time was measured in seconds with a hand stopwatch from the moment the pelvis was lifted off the mat to the moment the pelvis touched the mat. The endurance test of lateral trunk flexors was performed on both sides, first on the right and then on the left.

### 2.6. Rapid Movement Performance Tests

In the present study, we have selected tests that include typical rapid movements present during a football game: jumps, short sprints and changes of direction. The order of the tests was the same for all participants: standing long jump, 20 m sprint, Zig Zag change of direction test, and countermovement jump.

Standing long jump. At the beginning of the test, the participant stood next to the line mark on the floor and was instructed to jump as far as he could, using flexion in the knees and hips and swinging his arms in preparation. The distance between the starting mark and the landed heel was measured in meters after the jump was performed. The best result (the longest jump) of three repetitions was used for further analysis. Rest between each repetition was at least one minute. This test proved to be highly reliable and valid [20].

The 20 m sprint. This test was performed on an artificial grass and was measured using a pair of infrared photocells (Brower Timing System, Draper, UT, USA) placed at 20 m from the start at a hip-level height. At the beginning of the test, each participant took a high start position with the front foot close to the start line mark, and rear foot placed on a marked surface under which the foot switch sensor was placed. The instruction given to the participant was to run as fast as possible until he passed a pair of infrared photocells. The time was triggered when the participant started running, i.e., when he released the sensor from the pressure of the rear foot. For further analysis, the best result of two attempts was used, between which there were at least two minutes of rest.

Zig Zag test. According to Little et al. [21], Zig Zag test requires acceleration, deceleration, and a component of balance during running. The test was performed on artificial grass and was measured using two pairs of infrared stations (Brower Timing System, Draper, UT, USA) placed on the start and end line of the test field consisting of four 5 m sections placed at an angle of 100°. After explaining the test and instructing them to run between the markings as fast as they could, participants took a high start position and started the time by passing between the infrared station at the start and stopping the time by passing between the stations at the finish. For further analysis, the best result of two attempts is recorded, between which there were at least two minutes rest.

Countermovement jump. The OptoGait opto-electronic system (Microgate s.r.l., Bolzano, Italy) was used to measure countermovement jump height. At the beginning of the test, the participant took an upright standing position between two horizontally placed opto-electronic sensors. Hands were placed on the hips throughout the test performance. The participant was instructed to perform the highest possible jump preceded by the preparation in the form of a squat and to land at the same place from which he took off. Special attention was paid to avoid bending the legs at the hips, knees, and ankles when landing. The jump was repeated three times with a rest of at least one minute between repetitions, and for further analysis the best result (highest jump) was used.

### 2.7. Statistical Analysis

Descriptive statistics were calculated for all variables: arithmetic mean and standard deviation and normality of the distribution of variables was determined by the Kolmogorov-Smirnov (K-S) test. The inter-correlation matrix of all variables related to trunk function was factorized using principal components factor analysis (PCA). The data were analysed using a procedure called FACTOR in the SPSS (version 20.0) software package. The number of significant principal components in the factor pattern matrix extracted by the factor analysis was determined by the Kaiser–Guttman criterion, which retains principal components with eigenvalues greater than one. The original factor pattern matrix was rotated non-orthogonally to improve the simple structure of the matrix using Promax rotation [22]. This rotation also allowed us to evaluate inter-relations among extracted factors. The final outcomes of factor analysis were: eigenvalues and percentages of variance explained by each rotated factor, factor pattern matrix (partial standardized regression coefficients of each variable with a particular factor), factor structure matrix (zero-order correlations of each variable with a particular factor), and inter-correlations among the extracted factors. The associations between extracted factors representing trunk motor functions and rapid movement performance were determined using linear correlational analysis. Thresholds of 0.1, 0.3, 0.5, 0.7, and 0.9 for small, moderate, large, very large, and extremely large correlation coefficients were used [23]. The level of statistical significance was set at *p* < 0.05.

## 3. Results

Descriptive statistics and a test of normality of distribution of all variables representing trunk muscle function and rapid movement performance are shown in Table 1. Note that all trunk and rapid movement performance variables were normally distributed (K-S test < 0.14) except two trunk endurance tests (Table 1).

The applied PCA extracted four principal components that accounted for 70% of the variance of all variables related to trunk muscle function. Specifically, the first, second, third and fourth principal component explained 41%, 14%, 8% and 7% of the variance of all manifest variables. Table 2 presents factor pattern and structure matrices. The highest standardized regression coefficients (0.82 to 0.89) and correlations (0.81 to 0.91) with the first factor have all medicine ball throwing variables. The highest standardized regression coefficients (0.80 to 0.87) and correlations (0.71 to 0.88) with the second factor have all standing isometric strength tests. The highest standardized regression coefficients (0.47 to 0.69) and correlations (0.45 to 0.70) with the third factor have four trunk muscle endurance tests. Finally, the highest standardized regression coefficients (−0.63 to −0.85) and correlations (−0.53 to −0.84) with the fourth factor have all seated isometric strength tests. Based on these results the four extracted factors can be named as follows: (1) trunk (and upper body) power; (2) standing isometric trunk strength; (3) trunk muscle endurance; and (4) seated isometric trunk strength.

Table 3 presents inter-correlations among four extracted factors of trunk motor function. Note that the correlations among extracted factors were generally low (*r* = −0.26 to 0.42), suggesting that these four factors represent relatively independent trunk motor qualities.

The correlation coefficients between the extracted factors of trunk motor functions and rapid movement performance tests are shown in Table 4. In general, correlations were small to moderate, except for the first factor (i.e., trunk power), which had large, logically positive correlations with sprint (*r* = −0.5) and vertical jump performance (*r* = 0.6).

## 4. Discussion

To our knowledge, this is the first study that evaluated the factorial structure of trunk motor qualities and their associations with explosive muscular performance of athletes. The main findings of this study were as follows: (I) seventy percent of the structure of trunk motor qualities can be explained by the four independent principal components or factors—trunk power, standing isometric trunk strength, trunk muscle endurance, and seated isometric trunk strength; (II) trunk power had significant large positive associations with sprint and vertical jump performance, while other associations between trunk motor factors and explosive movement performance were generally small to moderate; (III) given that the standing and seated isometric trunk strength factors had small negative linear association, and only standing isometric trunk strength had significant, logically positive associations with explosive movement performance, seated trunk strength testing in football players can be considered as redundant.

### 4.1. Structure of Trunk Motor Qualities

In the present study, we have applied a PCA on a set of trunk power, strength and endurance variables to reveal the structure of trunk motor qualities in football players. The PCA revealed four independent principal components that explained 70% of the total variability of selected set of trunk motor variables, and that are relatively easy to interpret. The extracted components were further rotated using Promax non-orthogonal rotation, resulting in four relatively independent trunk motor factors.

The first factor was mainly loaded with trunk power variables, i.e., medicine ball throws. Notably, there were no differences in neither standardized regression coefficients nor correlation coefficients between static and dynamic medicine ball throws, and the first principal component (see Table 2). These results suggest that the first factor can be interpreted as trunk (and upper body) power, which is in line with our first hypothesis. Furthermore, our results strongly suggest that the researchers and practitioners can use either static or dynamic medicine ball tests for the assessment of trunk power, at least in male footballers.

The second factor was predominantly loaded with standing, but not seated, isometric trunk strength variables (Table 2). Instead, seated isometric trunk strength variables had the strongest parallel and orthogonal projections onto the fourth factor. Notably, linear correlation between the second and fourth factor was low and negative (see Table 3), suggesting that these two factors represent independent trunk strength qualities. This rather unexpected finding clearly indicates that the involvement of hip and lower body muscles strongly affects the strength potential of trunk muscles in sagittal and frontal plane. A plausible explanation could be found in mechanical force transmission between serially connected skeletal muscles within the myofascial chain [24]. For example, when considering posterior (or back) myofascial chain [25] weakness in the athletes’ hamstring or calf muscles is likely to affect their standing, but not seated, trunk extensor strength performance. Future studies are needed to further investigate this potentially highly relevant issue, at least from the perspective of testing and training of trunk muscle strength in sports. Overall, we may conclude that the second and fourth factors can be interpreted as standing and seated isometric trunk strength, respectively.

Finally, the strongest loadings on the third extracted factor had four trunk muscle endurance tests, previously improperly regarded as core stability tests [8,9,10,11,26]. This is again in line with our first hypothesis and suggests that the third factor can be interpreted as trunk muscle endurance. Of note is the fact that the correlation among the four selected trunk endurance tests (even between the left and right side-plank) was rather low (*r* < 0.25; data not shown). Previous studies also reported low to moderate linear associations among trunk muscle endurance tests [8,17].

Taken together, obtained structure of trunk motor qualities is in line with our first hypothesis, and with the definition of Silfies et al. [5] who considered trunk muscle strength, power and endurance as key motor qualities relevant for trunk stability.

### 4.2. Association between Trunk Motor Qualities and Explosive Movement Performance

The second important finding of the current study is related to generally low to moderate linear associations between the extracted trunk motor factors and performance in explosive movements. Only trunk power factor proved to be highly associated with sprint and vertical jump performance (see Table 4), with common variance ranging between 25% and 36%, respectively. These results support our second hypothesis, and are in line with some [11,16,27] but not all [8,12,28,29,30] previous research. Specifically, correlations between trunk power variables (static and dynamic medicine ball throws) and athletic performance variables (CMJ, 40 y dash and Agility Shuttle Run) in a study by Shinkle et al. [16] on football players range from 0.41 to 0.65. On the other hand, Nikolenko et al. [29] did not find significant correlations between trunk power and explosive movement performance. Notably, the authors assessed athletes’ trunk power using medicine ball throws from lying position, which could explain contradictory findings.

Our findings also suggest a weak association between trunk isometric strength and explosive movement performance in football players. Specifically, isometric trunk strength shared only about 10% of common variance with explosive movement performance tests. Clayton et al. [27] reported no association between the isokinetic strength of trunk flexors and extensors with vertical jump performance of sub-elite baseball players. Keiner et al. [29] reported moderate correlations (*r* = −0.45 to −0.48) between seated trunk strength and sprint performance of adolescent football players. Prieske et al. [12] found large correlations (*r* = 0.50–0.66) between isokinetic trunk extensor strength and jump performance, while no significant correlation was obtained for isokinetic strength of trunk flexors. The observed differences in results could be related to differences in sample studies, testing procedure (isometric vs. isokinetic tests) and in normalization of trunk strength for body size.

Finally, we found no or low correlations between trunk muscle endurance and explosive movement performance. Hoppe et al. [11] in a study on top hockey players also found no correlations between trunk muscular endurance tests and explosive movement performance. On the other hand, Okada et al. [28] found significant low to moderate correlations (−0.38 to −0.44) between endurance of trunk lateral flexors and change-of-direction speed in recreational males and females. The fact that authors included both sexes inevitably increased data variability, which commonly increases linear associations between independent variables. Nesser et al. [8] also obtained significant moderate correlations (0.38 to 0.49) between trunk muscular endurance tests and explosive movement performance of Division I Football players. Despite that, the authors concluded that core muscle function are not going to contribute to power performance and should not be the focus of strength and conditioning.

Obviously, more studies are needed to reveal the role of trunk motor functions in optimizing explosive movement performance in various sports.

### 4.3. Limitations

Like any study, this study is not without limitations. We see its main limitation in the generalizability of findings to populations other than young male football players. Hence, similar studies on other sports populations are needed to verify the validity of our findings. Another limitation of the current study is related to the fact that the results of some (e.g., trunk isometric strength), but not all (e.g., trunk isometric endurance), trunk muscle function tests were appropriate normalized for body size [31]. Indeed, it has been shown that muscle endurance tests in which individuals overcome their own body mass could penalize ones with larger body size [31]. Since the selected group of footballers were relatively homogenous with respect to body size (stature: 1.78 ± 0.07 m; body mass: 70.3 ± 7.5 kg) and having in mind that body size has generally lower effect on performance in muscle endurance vs. muscle strength tests [32], we believe that this limitation did not have fundamental effect on main findings of the current study.

## 5. Conclusions

Based on the results of the current study we may draw the following conclusions: (I) trunk muscle function of male football players can be described with three relatively independent trunk motor qualities: trunk power, trunk strength, and trunk endurance; (II) the structure of isometric trunk strength strongly depends on body position, with standing vs. seated testing position being more valid in the context of association between trunk strength and functional movement performance; (III) trunk power is positively associated with sprint and vertical jump performance, while other trunk motor qualities are generally unrelated with explosive movement performance in male football players.

From the practical point of view, our results suggest that coaches and practitioners should evaluate trunk muscle function of football players using medicine ball throws and isometric trunk strength tests, preferably in a standing position. Additional longitudinal training studies are needed to reveal the effects of specific trunk power training on explosive movement performance in football and similar sports.

## Figures and Tables

**Figure 1 sports-09-00067-f001:**
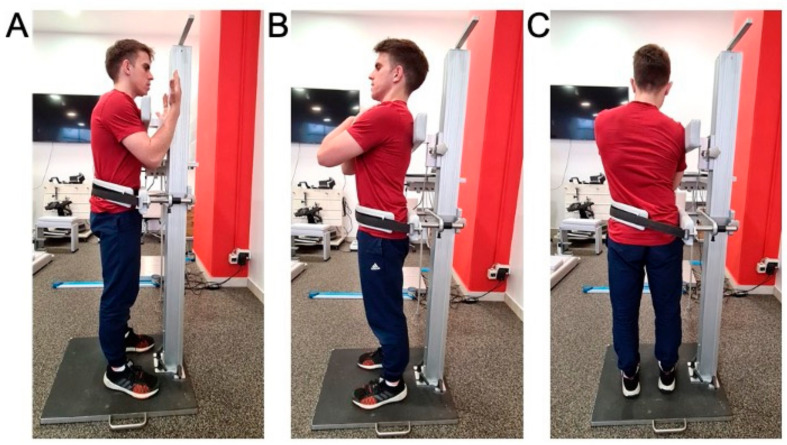
Isometric dynamometry of trunk extensors (**A**), trunk flexors (**B**) and trunk lateroflexors (**C**) in standing position.

**Figure 2 sports-09-00067-f002:**
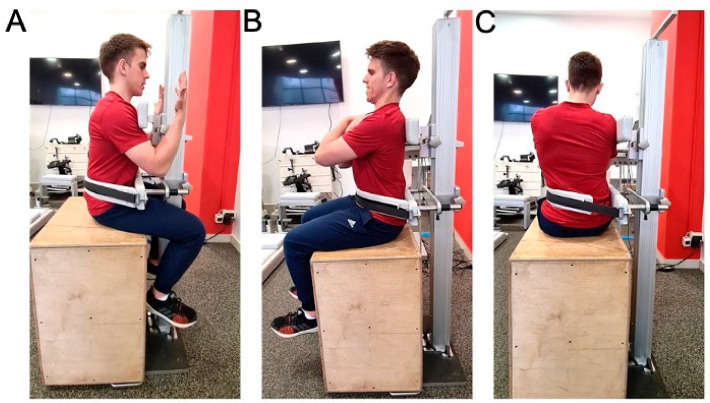
Isometric dynamometry of trunk extensors (**A**), trunk flexors (**B**) and trunk lateroflexors (**C**) in sitting position.

**Table 1 sports-09-00067-t001:** Descriptive statistics of all trunk and rapid movement performance tests.

	Mean	SD	K-S Test
Standing MVC trunk extension (Nm/kg)	4.32	1.08	0.06
Standing MVC trunk flexion (Nm/kg)	3.53	0.78	0.05
Standing MVC trunk lateroflexion—R (Nm/kg)	3.65	0.88	0.06
Standing MVC trunk lateroflexion—L (Nm/kg)	3.69	0.85	0.08
Seated MVC trunk extension (Nm/kg)	4.39	0.97	0.05
Seated MVC trunk flexion (Nm/kg)	2.35	0.46	0.06
Seated MVC trunk lateroflexion—R (Nm/kg)	2.87	0.65	0.07
Seated MVC trunk lateroflexion—L (Nm/kg)	3.07	0.68	0.05
Static forward throw (cm)	568.2	80.8	0.07
Static backward throw (cm)	579.9	78.8	0.05
Static side throw—right (cm)	552.9	82.9	0.05
Static side throw—left (cm)	568.7	83.2	0.08
Dynamic forward throw (cm)	590.4	78.7	0.05
Dynamic backward throw (cm)	759.0	108.0	0.11
Dynamic side throw—right (cm)	729.4	107.9	0.07
Dynamic side throw—left (cm)	746.0	113.2	0.06
Standing forward throw (cm)	789.9	125.9	0.05
Modified Biering–Sorensen endurance test (s)	168.8	59.5	0.13
V-sit endurance test (s)	224.2	138.6	0.18 *
Side plank endurance test—right (s)	128.7	39.9	0.10
Side plank endurance test—left (s)	141.8	117.2	0.27 *
Standing long jump (m)	2.35	0.18	0.05
Sprint 20 m (s)	3.09	0.16	0.13
Zig Zag test (s)	7.71	0.31	0.06
Countermovement jump (m)	0.36	0.05	0.09

Legend: MVC—maximum voluntary isometric action; Nm—Newton-meter; SD—standard deviation; K-S test—Kolmogorov–Smirnov test of normality of distribution; * significantly different from normal distribution.

**Table 2 sports-09-00067-t002:** Structure matrix and pattern matrix.

	Structure Matrix				Pattern Matrix			
	1	2	3	4	1	2	3	4
Stand. MVC trunk extension	−0.03	**0.87**	−0.09	−0.02	0.35	**0.86**	−0.03	−0.25
Stand. MVC trunk flexion	0.10	**0.71**	−0.02	−0.18	0.43	**0.80**	0.04	−0.39
Stand. MVC trunk lateroflexion R	0.04	**0.84**	0.05	−0.06	0.40	**0.87**	0.11	−0.29
Stand. MVC trunk lateroflexion L	−0.04	**0.88**	−0.06	−0.04	0.34	**0.87**	0.01	−0.26
Seat. MVC trunk extension	0.08	0.36	−0.25	**−0.53**	0.34	0.52	−0.21	**−0.63**
Seat. MVC trunk flexion	0.20	0.11	0.11	**−0.65**	0.36	0.37	0.13	**−0.72**
Seat. MVC trunk lateroflexion R	−0.07	0.06	0.15	**−0.84**	0.10	0.26	0.17	**−0.85**
Seat. MVC trunk lateroflexion L	0.04	0.37	0.25	**−0.60**	0.30	0.57	0.30	**−0.72**
Static forward throw	**0.81**	0.14	−0.06	0.02	**0.86**	0.47	−0.06	−0.16
Static backward throw	**0.85**	0.10	−0.09	0.09	**0.87**	0.42	−0.10	−0.09
Static side throw R	**0.88**	−0.01	0.02	−0.06	**0.89**	0.38	0.00	−0.22
Static side throw L	**0.91**	−0.06	−0.16	0.01	**0.88**	0.31	−0.18	−0.13
Dynamic forward throw	**0.88**	0.04	0.01	0.05	**0.88**	0.40	−0.00	−0.12
Dynamic backward throw	**0.80**	0.03	0.09	−0.04	**0.82**	0.38	0.07	−0.19
Dynamic side throw R	**0.91**	−0.12	0.07	0.03	**0.85**	0.26	0.05	−0.10
Dynamic side throw L	**0.91**	−0.10	0.01	−0.05	**0.87**	0.29	−0.01	−0.19
Standing forward throw	**0.88**	0.01	0.03	−0.02	**0.89**	0.39	0.02	−0.18
Modified Biering–Sorensen test	−0.26	−0.06	**0.57**	0.22	−0.34	−0.19	**0.56**	0.27
V-sit endurance test	−0.04	−0.20	**0.70**	−0.24	−0.09	−0.10	**0.69**	−0.20
Side plank R	0.13	0.11	**0.65**	−0.18	0.20	0.26	**0.66**	−0.25
Side plank L	0.16	0.40	**0.45**	0.41	0.24	0.40	**0.47**	0.26

Legend: MVC—maximum voluntary isometric action; R—right side; L—left side.

**Table 3 sports-09-00067-t003:** Intercorrelation matrix of extracted factors (* *p* < 0.05).

	1	2	3	4
**1**	1.00			
**2**	0.42 *	1.00		
**3**	−0.02	0.07	1.00	
**4**	−0.18	−0.26 *	−0.03	1.00

**Table 4 sports-09-00067-t004:** Correlation coefficients (Pearson’s r) between the four extracted factors of trunk motor qualities and rapid movement performance tests (* *p* < 0.05).

	Extracted Factors			
	1	2	3	4
Standing long jump	−0.26 *	−0.33 *	0.10	−0.30 *
Sprint 20 m	−0.50 *	−0.23	0.12	0.31 *
Zig Zag test	−0.12	−0.25 *	−0.15	0.43 *
Counter movement jump	0.60 *	0.29 *	0.18	−0.20

## Data Availability

Data sharing is not applicable to this article.

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
