# Peer review of "Factorial Structure of Trunk Motor Qualities and Their Association with Explosive Movement Performance in Young Footballers"

_sports, 2021, doi:10.3390/sports9050067_

Round 1
Reviewer 1 Report
Thank you for the opportunity to review the manuscript. Manuscript is well prepared and interesting, Sample of participants is of appropriate level and size. Conclusions are solid and backgrounded. I provided some comments and suggestions which will hopefully improve the readability of the paper.
Abstract
Use body mass instead of weight (in the whole manuscript)
Methods
More details are needed about the training status of the players (number of training sessions per week, distribution of the technical tactical vs. conditioning training, etc.).
I will strongly suggest authors to include figures of testing procedures. Otherwise, the tests are understandable only for those who are absolutely familiar with it. Sports is e-journal and you should not worry about number of pages, and as far as I'm aware there is no limitation in number of figures, tables, etc.
Results
I suppose you did at least several trials for each test. You should present intra-testing reliability
Tables needs legends
After you rotated originally obtained principal components, you should speak about "factors", and not "principal components" Principal components are originally ortogonal one to another (the 1st phase of the factor analysis extract principal components), and you are calculating correlations among them (Table 3), meaning that they are not ortogonal, and therefore and not "principal components". Accordingly, you should adapt the text of the manuscript (and title of the paper)
Author Response
Thank you for the opportunity to review the manuscript. Manuscript is well prepared and interesting, Sample of participants is of appropriate level and size. Conclusions are solid and backgrounded. I provided some comments and suggestions which will hopefully improve the readability of the paper.
Abstract
Use body mass instead of weight (in the whole manuscript)
A: As suggested, we have replaced the term „weight“ with the term “body mass”.
Methods
More details are needed about the training status of the players (number of training sessions per week, distribution of the technical tactical vs. conditioning training, etc.).
A: As suggested, we have added information related to the training status of tested players (line 87).
I will strongly suggest authors to include figures of testing procedures. Otherwise, the tests are understandable only for those who are absolutely familiar with it. Sports is e-journal and you should not worry about number of pages, and as far as I'm aware there is no limitation in number of figures, tables, etc.
A: We appreciate the reviewer’s suggestion. However, it is unrealistic to present all 25 tests in figures. After all, a number of tests in the present study have been used in numerous previous studies. We therefore added 2 figures in which we presented only tests that are rarely used in sports science research, i.e., standing and seated isometric MVC tests.
Results
I suppose you did at least several trials for each test. You should present intra-testing reliability.
A: In the present study, some tests (e.g., trunk endurance tests) have been performed only once, while for some tests we performed 3 trials. As explained in the Methods section, we have selected tests that proved to have high inter-session reliability in previous research. We, therefore, do not see much relevance of calculating intra-session reliability as it is generally higher (and less relevant) than the inter-session reliability.
Tables needs legends
A: We appreciate the suggestion. We therefore added legends for all tables that have abbreviations.
After you rotated originally obtained principal components, you should speak about "factors", and not "principal components" Principal components are originally ortogonal one to another (the 1st phase of the factor analysis extract principal components), and you are calculating correlations among them (Table 3), meaning that they are not ortogonal, and therefore and not "principal components". Accordingly, you should adapt the text of the manuscript (and title of the paper)
A: The reviewer is right; we have corrected this mistake in the whole manuscript. We have also added the word „Factorial“ in the title.
Reviewer 2 Report
Dear Authors,
Thank you for letting me revise your manuscript. Below you will find my suggestions.
L11: Please remove subheadings from the abstract.
L29-36: The authors do not need to explain what the trunk is, but only the relevant part of their investigation.
L84: Please provide standard deviations and mean for age. Also, stature is height.
L89: Please provide IRB number.
L280: How is the correlation coefficient assessed? 0-0.1: low, 0.1-0.3: moderate?
L292: There is a problem with Table 1. You showed mean and standard deviation and also, the p-value of the K-S test, indicating a lack of normal distribution (p<0.05) for most tests (at least we don’t know as you only showed two decimal places). If tests were normally distributed, the mean and standard deviation are fine. However, mean and standard deviation are only valuable to describe the central tendency and dispersion in normal quantitative variables and in a case that the variable is not normally distributed another indicator called median and interquartile range (IQR= Q3-Q1) should be used.
This leads me to the following question: is the principal components factor analysis (PCA) adequate for non-parametric distributions?
L316: How is possible to calculate Pearson’s r (for normal distributions) when you have indicated that your distribution is not normally distributed? I assume the rest of the correlation coefficients are Pearson’s, so they are not adequately calculated for your distribution.
Author Response
Thank you for letting me revise your manuscript. Below you will find my suggestions.
L11: Please remove subheadings from the abstract.
A: Removed as suggested.
L29-36: The authors do not need to explain what the trunk is, but only the relevant part of their investigation.
A: We appreciate the reviewer's suggestion; however, we have found inconsistencies in definition and anatomical description of the trunk (core) in scientific literature. We would, therefore, prefer to keep this short paragraph in the Introduction section.
L84: Please provide standard deviations and mean for age. Also, stature is height.
A: We have added requested data, and we replaced the term “stature” with height.
L89: Please provide IRB number.
A: Added as requested.
L280: How is the correlation coefficient assessed? 0-0.1: low, 0.1-0.3: moderate?
A: The description of the thresholds for interpretation of the correlation coefficients is added at the end of the Methods section. Accordingly, we have modified the whole manuscript.
L292: There is a problem with Table 1. You showed mean and standard deviation and also, the p-value of the K-S test, indicating a lack of normal distribution (p<0.05) for most tests (at least we don’t know as you only showed two decimal places). If tests were normally distributed, the mean and standard deviation are fine. However, mean and standard deviation are only valuable to describe the central tendency and dispersion in normal quantitative variables and in a case that the variable is not normally distributed another indicator called median and interquartile range (IQR= Q3-Q1) should be used.
A: We appreciate the reviewer’s comment. The last column of Table 1 presents actual K-S test values (D), not its p-values, as the reviewer suggests. We have calculated again the K-S test for all variables. The critical value for our study is 0.14, so all but two (two muscle endurance tests) variables have normal distribution. We have corrected that mistake in the text and Table 1.
This leads me to the following question: is the principal components factor analysis (PCA) adequate for non-parametric distributions?
A: as explained above, the distribution of virtually all variables was normal so calculation of principal component factor analysis was appropriate.
L316: How is possible to calculate Pearson’s r (for normal distributions) when you have indicated that your distribution is not normally distributed? I assume the rest of the correlation coefficients are Pearson’s, so they are not adequately calculated for your distribution.
A: The distribution of all explosive performance variables was normal, so it is methodologically justified to calculate Pearson’s r between these variables and extracted factors.
Round 2
Reviewer 2 Report
Dear Authors,
Thank you for all your positive answers. However, in my first review, I showed concern about K-S tests for normality. You replied that Table 1 was not showing p, but the test statistic. The aim of the K-S test is to demonstrate normality and you claimed that 0.14 was a proper test statistic for that. As an external reviewer and also for readers, this test statistic relies on the data, so 0.14 might be good for you, but not for other studies. Therefore, I suggest you to show the p-value, rather than the test statistic. With the p-value, everyone can check the normality of your data.
Author Response
Dear reviewer,
Thank you for you response to our revision. Honestly, due to the well known central limit theorem, your concerns are basically unjustified. Namely, we have calculated K-S test for selected variables as this study is part of the first author's Ph.D. thesis, but K-S test is generally robust against a violation of assumption of normal distribution if sample sizes are reasonable, say N ≥ 25. Furthermore, newer versions of SPSS program do not calculate the exact p values for KS test statistic D but only asymptomatic values or lower bound of the true significance. What this means that if we include this statistic in table 1, you would see for different D values for K-S test the same asymptomatic p-value of 0.2. I do not see the point of adding this column to the table as it gives no additional information to the reader compared with what we have now. I hope that the the reviewer will accept this explanation as valid reason of not adding additional column with the same p-values for all variables except one that violates the assumption of normality.